# Exploring the Link between Varicella-Zoster Virus, Autoimmune Diseases, and the Role of Recombinant Zoster Vaccine

**DOI:** 10.3390/biom14070739

**Published:** 2024-06-22

**Authors:** Ryuhei Ishihara, Ryu Watanabe, Mayu Shiomi, Masao Katsushima, Kazuo Fukumoto, Shinsuke Yamada, Tadashi Okano, Motomu Hashimoto

**Affiliations:** 1Department of Clinical Immunology, Osaka Metropolitan University Graduate School of Medicine, Osaka 545-8585, Japan; 2Center for Senile Degenerative Disorders (CSDD), Osaka Metropolitan University Graduate School of Medicine, Osaka 545-8585, Japan

**Keywords:** autoimmune diseases, herpes zoster, Janus kinase inhibitors, recombinant zoster vaccine

## Abstract

The varicella-zoster virus (VZV) is a human neurotropic herpes virus responsible for varicella and herpes zoster (HZ). Following primary infection in childhood, VZV manifests as varicella (chickenpox) and enters a period of latency within the dorsal root ganglion. A compromised cellular immune response due to aging or immunosuppression triggers viral reactivation and the development of HZ (shingles). Patients with autoimmune diseases have a higher risk of developing HZ owing to the immunodeficiency associated with the disease itself and/or the use of immunosuppressive agents. The introduction of new immunosuppressive agents with unique mechanisms has expanded the treatment options for autoimmune diseases but has also increased the risk of HZ. Specifically, Janus kinase (JAK) inhibitors and anifrolumab have raised concerns regarding HZ. Despite treatment advances, a substantial number of patients suffer from complications such as postherpetic neuralgia for prolonged periods. The adjuvanted recombinant zoster vaccine (RZV) is considered safe and effective even in immunocompromised patients. The widespread adoption of RZV may reduce the health and socioeconomic burdens of HZ patients. This review covers the link between VZV and autoimmune diseases, assesses the risk of HZ associated with immunosuppressant use, and discusses the benefits and risks of using RZV in patients with autoimmune diseases.

## 1. Introduction

Varicella-zoster virus (VZV) is an enveloped virus containing double-stranded DNA and belongs to the alpha herpesvirus family. It is highly cell-associated and exclusively infects human cells, such as epithelial cells, T lymphocytes, and ganglion neurons [1]. Alpha herpesviruses possess numerous genes encoding glycoproteins, such as glycoprotein B (gB), gC, gD, gE, gI, gH, gL, gK, gM, and gN; however, VZV lacks genes encoding gD homologs. Unlike other herpesviruses, gE is one of the most abundant glycoproteins in VZV and is indispensable for growth [2].

Primary VZV infection results in a varicella (chickenpox) infection. Varicella is one of the most contagious human diseases and is transmitted via airborne routes (Figure 1) [3]. Initial viral replication occurs in the upper respiratory tract, leading to a vesicular rash covering the entire body after an incubation period of 10–21 days. This process reflects the replication of the virus in the tonsils and regional lymphoid tissue, where infected T lymphocytes carry the virus to the skin via the bloodstream [2]. T lymphocytes transport the virus to the skin and facilitate the VZV infection of epidermal cells. Epidermal cells adjacent to VZV-infected cells exhibit heightened interferon (IFN)-α production, which could potentially contribute to the control of VZV pathogenicity [4]. During this period, the virion travels retrogradely from the site of skin replication to the nerve cell body of a sensory ganglion neuron, establishing a latent infection. Cellular immunity subsequently regulates VZV replication, and viral reactivation remains subclinical and asymptomatic. Subclinical VZV replication contributes to an enhanced VZV-specific immune response in the host. However, herpes zoster (HZ) develops with declining cellular immunity due to aging or immunodeficiency. VZV reaches the skin through anterograde axonal transport, resulting in a vesicular rash on skin segments innervated by the affected ganglia (Figure 1) [5].

HZ manifests as a spectrum of clinical symptoms and complications, including postherpetic neuralgia (PHN), ocular HZ, meningitis, bladder and bowel dysfunction, and Ramsay Hunt syndrome. HZ is non-seasonal and occurs worldwide. A report from the United States indicates that HZ affects 1.2–3.4 cases per 1000 persons per year among young adults, in contrast to 3.9–11.8 cases per 1000 persons per year among individuals aged ≥65 years [6]. The heightened susceptibility among older adults may be attributed to immunosenescence, with a 50% probability of HZ in unvaccinated individuals surviving for 85 years [7]. Recently, the incidence of HZ has increased because of an increase in the number of older adults, immunosuppressed organ transplant patients, patients receiving chemotherapy for cancer or autoimmune diseases, individuals infected with the human immunodeficiency virus (HIV), and patients with chronic diseases [8]. Additionally, there have been some interesting reports of VZV reactivation following COVID-19 infection and vaccination [9,10]. The incidence of complications associated with HZ is higher in immunocompromised individuals than in immunocompetent individuals [11]. Studies suggest that the incidence of HZ events is elevated in individuals with autoimmune diseases such as rheumatoid arthritis (RA) [12], systemic lupus erythematosus (SLE) [13], Sjögren’s syndrome (SS) [14], psoriasis [15], inflammatory bowel disease (IBD) [16], and idiopathic inflammatory myopathy (IIM) [17]. This review illustrates the association between VZV infection and autoimmune diseases, with a focus on rheumatoid arthritis, systemic lupus erythematosus, and vasculitis. We assessed the risk of developing HZ when using immunosuppressants. Finally, we discuss the effectiveness and risks of the adjuvanted recombinant zoster vaccine (RZV, Shingrix^®^) against patients with autoimmune diseases.

## 2. Intersection of VZV and Autoimmune Diseases

Autoimmune disorders result from a disruption in immune tolerance, wherein the immune system fails to effectively differentiate between self and non-self [18]. Both genetic and environmental influences are believed to contribute significantly to the onset of autoimmune conditions, with viral infections representing one of the environmental factors that precipitate these disorders [19]. In recent years, COVID-19 has been reported to induce various autoimmune diseases, bringing renewed attention to the role of viral infections as triggers for the development of autoimmune conditions [20]. Furthermore, patients with autoimmune diseases face an elevated risk of infections, including HZ, owing to the immunodeficiency associated with the disease itself and/or the use of immunosuppressive agents. Even those <50 years of age have a 1.5- to 2-fold higher risk of HZ compared to healthy older adults [21,22].

Patients with autoimmune diseases frequently have comorbidities such as hypertension, dyslipidemia, obesity, diabetes, cardiovascular disease, osteoporosis, depression, and interstitial lung disease [23]. Reports indicate that hypertension and dyslipidemia increase the risk of HZ in RA patients [24], whereas diabetes does not appear to be associated with the risk of HZ in RA and SLE patients [22,24]. However, the evidence regarding the association between comorbidities and the development of HZ in patients with autoimmune diseases remains very limited.

Here, we delineate the intersections between autoimmune diseases and VZV, focusing on rheumatoid arthritis, systemic lupus erythematosus, and vasculitis.

### 2.1. Rheumatoid Arthritis

RA is a chronic inflammatory joint disease, most commonly targeting the small joints of the hands and feet, thus impacting all aspects of daily life [25]. Elevated levels of inflammation are associated with progressive disability, premature mortality, and substantial socioeconomic costs. It is estimated that approximately 17.6 million people have RA worldwide [26].

The environmental risk factors for RA include smoking, periodontal disease, and the microbiome, all of which interact with mucosal surfaces such as the lungs, oral mucosa, and gastrointestinal mucosa [27]. Respiratory viruses commonly affect both the oral mucosa and lungs; therefore, these local stresses may contribute to the development of RA. Indeed, the literature indicates that infections with parainfluenza viruses, coronaviruses, and metapneumoviruses are linked to an increased incidence of RA [28]. VZV can also be transmitted through contact with respiratory droplets and airway mucosa. However, the association between HZ and the onset or exacerbation of RA remains unclear. Paradoxically, there are documented cases of juvenile RA remission following varicella infection, as well as instances of RA remission after HZ; however, the underlying mechanism behind these phenomena has not been fully elucidated [29,30,31].

The risk of HZ in patients with RA is influenced by medication. In the United States, the incidence of HZ in patients with RA was 12.1 cases per 1000 person-years, compared to 5.4 cases in healthy controls [12]. In Japan, the incidence of HZ was similarly elevated in patients with RA, at 9.1 cases per 1000 person-years compared with 4.15 cases in the general population [32]. The risk of HZ attributed to RA is estimated to be approximately twice that of the general population [12,33].

### 2.2. Systemic Lupus Erythematosus

SLE is a multisystem autoimmune disorder characterized by the breakdown of self-tolerance and the activation of autoreactive T and B cells, leading to tissue injury through immune complex deposition and antibody production [34]. The most common clinical features of SLE include mucocutaneous, musculoskeletal, hematological, and renal involvement [35]. The disease predominantly affects females, with a ratio of approximately nine women to one man, and primarily affects individuals aged 15–45 years. It is estimated that SLE affects approximately 3.41 million people worldwide [36]. While several viruses, such as the Epstein–Barr virus and parvovirus B19, have been implicated in the onset and disease flares of SLE [37], there are few reports suggesting an association between HZ and SLE disease onset or flares [38,39].

In contrast, HZ is commonly observed in patients with SLE, with an incidence ranging from 6.4 to 91.4 cases per 1000 person-years [14,20,40,41,42]. The highest incidence is reported in Asian countries, particularly in Japan, where an incidence rate ranges from 53.7 to 91.4 per 1000 person-years [14,42]. The prevalence of HZ in Japanese patients with SLE is 43% [32], which is significantly higher than the 5–6% prevalence in Western populations [20,38]. Despite the high incidence, patients with SLE appear to have fewer severe HZ-related complications and mortality rates [38,40]. Furthermore, Francesco et al. reported that the incidence of HZ did not increase during the active phase of SLE, with approximately half of the patients developing HZ during the remission phase of lupus nephritis (LN). Moreover, approximately a quarter of these patients did not undergo immunosuppressive therapy. These findings suggest that factors beyond immunosuppressive therapy may contribute to the development of HZ in patients with SLE [39].

Patients with SLE exhibit a higher incidence of HZ despite having elevated IgG VZV antibody levels than healthy controls. In contrast, the frequency of virus-specific IFN-γ- or tumor necrosis factor (TNF)-α-positive CD4+ T cells is reduced in SLE patients compared to healthy controls and those with RA. Thus, the increased risk of HZ in patients with SLE may be associated with impaired CD4+ T-cell responses [43,44].

### 2.3. Vasculitis

Systemic vasculitis refers to a group of rare diseases characterized by inflammation of blood vessels, leading to ischemia and organ damage [45]. Vasculitis is defined by the size of the vessels involved: large vessels include the aorta and its major branches; medium vessels encompass the main visceral arteries and their branches; and small vessels include intraparenchymal arteries, arterioles, capillaries, and venules [46].

VZV is the only virus that replicates in the arteries and causes vascular damage in humans [47]. Historically, VZV vasculopathy has been delineated by its impact on intracranial arteries, often presenting as a transient ischemic attack or ischemic or hemorrhagic stroke [48]. Interleukin (IL)-8, IL-6, matrix metalloproteinase (MMP)-2, and MMP-9 levels were elevated in the cerebrospinal fluid of the patients with VZV vasculopathy (Figure 2). IL-8 acts as a chemoattractant for neutrophils, while IL-6 promotes macrophage differentiation. Elevated levels of proteases such as MMP-2 and MMP-9 disrupt basement membranes and tight junctions between endothelial cells, potentially contributing to inflammation and vessel wall damage, which are characteristics of VZV vasculopathy [49,50]. Macrophages have been suggested to be a source of MMP in murine ischemic models [51]. Additionally, programmed death-ligand 1 (PD-L1) expression was downregulated in VZV-infected adventitial cells in vitro, implying that the downregulation of PD-L1 may explain its ability to promote persistent inflammation in infected arteries [52]. Notably, not all patients with VZV angiopathy have a history of HZ or varicella rashes. In individuals presenting with unexplained unifocal or multifocal vasculopathy of the central nervous system, the CSF should be analyzed for the presence of VZV DNA and anti-VZV IgG antibodies [52].

VZV may trigger giant cell arteritis (GCA), an autoimmune vasculitis that affects medium- and large-sized vessels [53]. The prevalence of GCA in patients with HZ was elevated compared to that in the general population (0.340% vs. 0.143%) [54]. Some studies have demonstrated that VZV antigen was detected in approximately 75% of temporal artery specimens from patients histologically diagnosed with GCA [55]. Moreover, giant cells were observed in approximately 90% of the sections adjacent to the VZV antigen. VZV antigens were predominantly found in the adventitia, followed by the tunica media and intima [55]. In cases of GCA associated with VZV infection, some researchers advocate the use of long-term antiviral medications (such as valacyclovir) in combination with glucocorticoid therapy [56]. In GCA, IL-6, MMP-2, and MMP-9 have been reported to be critical for pathogenesis, along with a decrease in PD-L1 expression by dendritic cells in the adventitia [57,58,59]. Thus, there are many similarities between the pathogenesis of VZV angiopathy and GCA. However, many reports have suggested that GCA and VZV are not related, making this association controversial [60,61,62,63,64].

The association of VZV with vasculitis other than GCA remains unclear, with only sporadic case reports indicating that varicella/HZ is a potential trigger for the onset of ANCA-associated vasculitis or disease activity flareups [65,66]. Moreover, patients with ANCA-associated vasculitis did not exhibit differences in their VZV antibody levels or cellular immune response compared to healthy controls [44].

## 3. Impact of Immunosuppressive Therapy on VZV Reactivation

Disease-modifying antirheumatic drugs (DMARDs) and glucocorticoids (GCs) are used to control disease activity and achieve remission in autoimmune diseases such as RA and SLE. GCs have been used in the treatment of autoimmune diseases for more than 70 years, and despite their diverse array of side effects, they continue to be indispensable in routine clinical practice. DMARDs, including conventional synthetic, biological, and targeted synthetic DMARDs, are being increasingly utilized for numerous indications, thereby exposing a growing number of patients to the risk of infections [67,68]. However, evidence supporting the prevalence and management of these infections is limited.

### 3.1. Glucocorticoids

GCs induce lymphocyte apoptosis and act simultaneously on a wide range of cells involved in innate and acquired immunity, thereby exerting anti-inflammatory effects [69]. Therefore, GCs are employed in the treatment of various autoimmune diseases, but they also lead to numerous side effects, including infection, weight gain, osteoporosis, metabolic abnormalities, cataracts, and psychological stress [70]. It is also known to increase the risk of infection in a dose-dependent manner [71].

Results from a large cohort study revealed that systemic GC users have a 59% higher risk of HZ, with the risk escalating at higher cumulative doses. For doses <500, 500–1000, and ≥1000 mg of prednisolone equivalents, the hazard ratios were 1.32 (95% CI: 1.17–1.48), 1.74 (95% CI: 1.55–1.95), and 1.80 (95% CI: 1.61–2.02), respectively. Compared to non-users of GCs, the risk of HZ increased 1 month after a single prescription of systemic GCs [72]. The odds ratio increased by a 1.13 per 1 mg increase in prednisolone dose and nearly doubled for doses of 10 mg or more [73,74,75].

### 3.2. Conventional Synthetic DMARDs (csDMARDs)

DMARDs are therapeutic agents for the management of inflammatory arthritis such as RA, psoriatic arthritis (PsA), and ankylosing spondylitis. Additionally, they are utilized in the treatment of connective tissue disorders, including systemic sclerosis, SLE, and SS, as well as other conditions such as IIM, vasculitis, uveitis, and IBD. Each DMARD has a distinct mechanism of action that ultimately inhibits critical pathways of the inflammatory cascade. Data on the correlation between csDMARDs and infections in patients with RA are limited [76]. Some studies demonstrated that methotrexate (MTX), salazosulfapyridine, and leflunomide do not increase the risk of HZ [73,77,78]. However, hydroxychloroquine (HCQ) and azathioprine (AZP) are associated with HZ risk. The odds ratio for HCQ ranges from 1.77 to 1.95, suggesting that it may be an independent risk factor in studies of patients with RA [12,74,79]. The odds ratios for AZP ranged from 1.57 to 2.0 [77,80]. A network meta-analysis involving patients with lupus nephritis revealed that tacrolimus is associated with an increased risk of HZ [81].

The administration of cyclophosphamide or mycophenolate mofetil (MMF) presents a clear risk for HZ. Studies report a hazard ratio for cyclophosphamide ranging from 2.69 to 4.2, with a higher risk associated with oral rather than intravenous administration [77,79,80]. MMF has been used as an anchor drug for lupus nephritis, displaying higher efficacy and a reduced incidence of adverse events, including infection, compared with oral cyclophosphamide [82,83]. Nevertheless, studies involving patients with lupus nephritis and organ transplant recipients revealed an HZ risk of approximately twice that of GCs and AZP [40,81].

### 3.3. Biologic DMARDs (bDMARDs)

The use of bDMARDs as an adjunctive therapy to csDMARDs in the management of autoimmune and rheumatic diseases is rapidly expanding owing to the advantageous efficacy and safety profile of these agents. The principal targets of most bDMARDs include cytokines, B cells, and costimulatory molecules. bDMARDs have an elevated risk of infection compared to csDMARDs [84]. Patients receiving bDMARDs either as monotherapy or in combination with csDMARDs had a higher risk of HZ in comparison to those treated solely with csDMARDs (adjusted hazard ratio [aHR]: 5.53 [95% confidence interval {CI}: 2.03–3.16] vs. csDMARD aHR: 1.48 [95% CI: 1.33–1.66]) [77].

A retrospective cohort study involving patients with RA demonstrated no disparity between TNF inhibitors (TNFis), IL-6 inhibitors (IL-6i), and T-cell costimulation modulators [85]. In general, TNFis carry approximately double the risk of HZ compared to csDMARDs. However, the risk was diminished for soluble TNF receptors compared to anti-TNF monoclonal antibodies. This variance in risk may be explained by the mechanism by which anti-TNF monoclonal antibodies cross-link transmembrane TNF and prompt T-cell apoptosis, whereas the soluble TNF receptor lacks this activity [73,86,87,88,89,90]. In patients with RA, the odds ratio for rituximab-induced HZ was 2.06 [91].

Anifrolumab is a monoclonal antibody against the type I interferon receptor [92]. Activation of the type I IFN system is considered to be a central pathogenic mediator in SLE [93]. Cell signaling by all type I IFNs, including IFNα and IFNβ, is mediated by the type I IFN receptor, which activates the Janus kinase (JAK)–signal transducer and activator of transcription (STAT) pathway, leading to the transcription of IFN-stimulated genes [94]. Consequently, blocking the type I IFN receptor is one of the emerging therapeutic strategies for SLE [95]. In 2021, intravenous anifrolumab was approved in the United States for the treatment of adults with moderate-to-severe SLE who were on standard therapy [96]. Type I IFN signaling not only plays a pivotal role in SLE activity but also constitutes the central mechanism of antiviral immunity [97]. Hence, it is not surprising that clinical trials of anifrolumab have revealed a drug-associated risk of HZ. In the Treatment of Uncontrolled Lupus via the Interferon Pathway (TULIP) trial and the 3-year long-term extension study, which compared anifrolumab 300 mg with placebo in patients with SLE, the incidence of HZ was 13.4% in the anifrolumab group and 3.6% in the placebo group over the entire period. Data from a subgroup analysis of Japanese patients revealed that the incidence of HZ was 24.2% and 5.3% in the anifrolumab and placebo groups, respectively. The results for the overall population indicated a decline in the incidence of HZ over time among patients treated with anifrolumab [98,99].

### 3.4. Targeted Synthetic DMARDs (tsDMARDs)

JAKs are a family of cytoplasmic non-receptor tyrosine kinases comprising four members: JAK1, JAK2, JAK3, and TYK2. JAKs transduce cytokine signaling through the JAK-STAT pathway and oversee the transcription of numerous genes involved in inflammation, immunity, and cancer [100]. JAK inhibitors (JAKis) are a relatively recent class of immunosuppressive agents that target intracellular signaling pathways pivotal in the pathogenesis of autoimmune diseases [101]. In 2012, tofacitinib was the first JAKi approved for RA treatment. Since then, the indications of JAK inhibitors have expanded to other autoimmune diseases, such as PsA and IBD [102]. Recent evidence suggests that JAKis may be effective in the treatment of large-vessel vasculitis, including Takayasu arteritis and GCA [103]. However, the use of JAKis is linked to the risk of severe infections, particularly HZ. T helper type 1 (Th1) cells primarily induce adaptive responses against intracellular pathogens, such as viruses. The cytokines necessary for the differentiation into Th1 cells are IFN-γ and IL-12, whose intracellular pathways are mediated by JAK1-2/STAT1 and JAK2-TYK2/STAT4, respectively. IFN signaling directly inhibits intracellular viral replication, enhances antigen presentation, and induces the differentiation of CD8+ effector cells. JAK1 and JAK3 are essential for effective antibody production and B-cell differentiation [104,105,106]. Thus, with JAKis, which block these signaling pathways, caution must be exercised regarding viral infections, particularly those caused by the herpes virus.

Data from the upadacitinib (UPA) clinical trial revealed that the incidence of HZ per 100 patient-years was 0.8, 1.1, 3.0, and 5.3 in the MTX alone, adalimumab + MTX, UPA 15 mg, and UPA 30 mg groups, respectively. Notably, a history of HZ and residence in the Asian region were identified as risk factors for the onset of HZ [107]. This effect of ethnicity on the risk of developing HZ in RA patients was also observed with tofacitinib [108]. Similarly, other JAKis exhibit an elevated risk of developing HZ compared to csDMARDs and bDMARDs. HZ is regarded as a class effect of JAKis, with a potential dose-dependent escalation in risk [109,110,111]. Findings from a recent network meta-analysis indicated that the risk of HZ might be reduced with filgotinib compared with UPA and baricitinib [112,113].

Thus, csDMARDs, bDMARDs, and tsDMARDs carry a high risk of HZ, and all physicians who prescribe immunosuppressive agents, not just rheumatologists, should be aware of this risk.

## 4. Vaccines in the Context of Autoimmune Diseases

There are over 1 million cases of HZ in the United States annually, with an annual incidence rate of 3–4 cases per 1000 people [6]. Following the introduction of acyclovir and other anti-herpes virus medications, the treatment outcomes for HZ have markedly improved [114,115,116]. Nonetheless, numerous patients continue to endure various complications and PHN for extended periods, which constitutes a significant global public health burden. For instance, in Latin America and the Caribbean, hospitalization rates for HZ vary from 3% to 35.7%, with in-hospital mortality rates of up to 36% [11]. In the United States, HZ and its associated complications contribute $2.4 billion in direct medical costs and lost productivity annually [117].

In addition, HZ recurrence is also an issue. For instance, the recurrence rate of HZ after 8 years was 6.2%, with immunocompromised patients experiencing a recurrence rate 2.4 times higher than that of healthy controls [118]. In the general population, the risk of HZ recurrence was similar to the risk of the first episode of HZ, indicating that HZ is not a once-in-a-lifetime event and that experiencing HZ does not confer immunity against subsequent episodes. Thus, individuals even with a history of HZ should be advised to receive an HZ vaccine to prevent recurrence [118].

Currently, two main vaccine formulations are used: a live-attenuated zoster vaccine (ZVL, Zostavax; Merck, Darmstadt, Germany) and an adjuvanted recombinant zoster vaccine (RZV, Shingrix; GlaxoSmithKline, London UK). These vaccines can reduce the risks of HZ and PHN by 50–90% [119,120,121,122].

### 4.1. Live-Attenuated Zoster Vaccine

In Japan, in the early 1970s, Takahashi et al. developed a live-attenuated VZV (Oka strain) vaccine derived from the wild-type VZV. The Oka strain was attenuated through sequential passaging of the wild-type virus in human embryo fibroblast cells (MRC-5 cells) and guinea pig embryo fibroblasts [123]. In 1995, ZVL was approved in the United States, and routine vaccination at the age of 1 year was recommended for healthy children. In 2007, it was established that a second dose provided greater protection than a single dose [124]. This vaccination strategy reduced the occurrence of HZ and PHN by 51.3% and 66.5%, respectively [121]. Nevertheless, the effectiveness of ZVL may diminish with advancing age, exhibiting lower efficacy in individuals >70 years of age than in those aged 60–69 years (64% vs. 38%). This observation reflects a significantly diminished VZV-specific cellular immune response in individuals >70 years [125].

The efficacy of ZVL decreased from 67% at 1 year to 15% at 10 years of age. Similarly, the vaccine efficacy against PHN decreased from 83% to 41% after 10 years [126]. A booster vaccination administered to 200 participants aged ≥70 years who had received ZVL more than 10 years earlier demonstrated a greater VZV-specific cellular immune response than the initial dose group. Furthermore, this enhanced response was shown to persist for at least as long as the first dose group [127,128]. VZV booster vaccination may be beneficial in suppressing viral reactivation.

Clinical trials on ZVL have not included immunocompromised individuals [121,129]. As it is a live vaccine, the administration of ZVL is generally contraindicated in severely immunocompromised individuals. However, several studies have reported on the safety and efficacy of live vaccine administration in immunocompromised individuals. A randomized controlled trial involving 617 patients receiving TNFis demonstrated the safety and efficacy of this live-attenuated vaccine in this particular patient cohort [130,131]. ZVL was also effective and safe in patients with SLE [132]. A randomized controlled trial conducted in patients with HIV also observed no incidence of HZ after ZVL inoculation [133]. These findings indicate its safety and efficacy in immunocompromised patients. However, the vaccine is less effective in this population than in healthy individuals [134].

Conversely, ZVL inoculation leading to HZ or disseminated HZ has been reported [129]. The virus persists in the patient’s body for several weeks post-vaccination, with a small number of patients potentially shedding the virus in their saliva for up to 4 weeks following vaccination [135]. Hence, a period of approximately 4 weeks between vaccination and commencement of immunosuppressive treatment is recommended [126].

ZVL contains residual components of MRC-5 cells, including DNA and proteins, which can theoretically induce autoimmunity. Furthermore, as a DNA vaccine, it may contain antigens that can cross-react with autoantigens. The likelihood of arthritis was notably elevated in patients who received ZVL compared with those who received the tetanus toxoid-containing vaccine. Patients who received ZVL were 2.7 times more likely to develop arthritis [136].

### 4.2. Recombinant Zoster Vaccine

The adjuvanted recombinant zoster vaccine (RZV) is a novel zoster vaccine containing the VZV gE antigen, a primary target of CD4+ T-cell responses, and a liposome-based AS01B adjuvant (Figure 3) [137]. VZV gE is the most abundant viral glycoprotein and plays an indispensable role in the assembly of infectious virions and viral replication [138]. Moreover, gE is recognized as the primary target of VZV-specific CD4+ T-cell responses [139]. The AS01B adjuvant system comprises two immunostimulants. One is monophosphoryl lipid A, a detoxified derivative of lipopolysaccharide from Salmonella species, which stimulates nuclear factor-κB (NF-κB) transcription and cytokine production. As a Toll-like receptor (TLR) 4 agonist, it activates antigen-presenting cells. The second component is QS-21, a natural saponin extracted from Chilean soapbark trees. QS-21 enhances CD4+ T-cell-mediated immune response and antibody production [140]. RZV demonstrates greater efficacy against HZ than ZVL because of the potent gE adjuvanted with AS01B, which robustly induces gE-specific CD4+ T-cell and humoral immune responses [137]. This vaccine is usually administered in two doses spaced 2–6 months apart [122]. Two clinical trials, ZOE-50 (NCT01165177) and ZOE-70 (NCT01165229), were conducted in healthy participants aged ≥50 years and ≥70 years. The efficacy rates at 1 year were 97% and 89%, respectively, suggesting that the efficacy of the vaccine may be slightly reduced in older adults [122,137].

At 6 years after two RZV vaccinations, gE-specific cell-mediated immune responses were 3.8-fold higher than pre-vaccination levels, and anti-gE antibody levels were 7.3-fold higher than pre-vaccination levels, persisting at similar levels at the 9-year mark [141,142]. The median CD4+ T-cell and anti-gE antibody levels at 10 years were 3.3 and 5.9 times higher than pre-vaccination levels, respectively [142]. An interim analysis of this follow-up study of the ZOE-50 and ZOE-70 trials demonstrated vaccine efficacy for up to 10 years post-vaccination. The overall vaccine efficacy against HZ for immunocompetent individuals aged ≥50 years is approximately 90% during a mean follow-up of 9.6 years. However, vaccine efficacy gradually declined from 97% in the first year to approximately 73% in years 9 and 10 [143]. This sustained immune response is anticipated to remain as strong as the initial immune response above baseline for up to 15 years post-vaccination [142,144]. Moreover, RZV reduces the duration of HZ-associated pain and enhances patient quality of life by reducing the need for analgesics, even in breakthrough cases [145]. Thus, RZV vaccination not only reduces the incidence of HZ but also its complications and the utilization of medical resources, potentially leading to a considerable reduction in the public health burden [146].

In a pooled analysis of the ZOE-50 and ZOE-70 trials, RZV was associated with more adverse events than the placebo [147]. The most common adverse event was injection site pain (RZV: 78.0%, placebo: 10.9%), with Grade 3 pain occurring in 6.4% of RZV recipients and 0.3% of placebo recipients. Myalgia, fatigue, and headache were also frequently reported (RZV: 44.7%, 44.5%, and 37.7%; placebo: 11.7%, 16.5%, and 15.5%, respectively). Most symptoms were mild to moderate, with a median duration of 2–3 days. A pooled analysis of the primary phase III trials showed no clinically relevant differences between RZV and placebo in the overall incidence of serious adverse events, fatal adverse events, and potential immune-mediated diseases [147].

In a study investigating the impact of RVZ in older patients who had received ZVL more than 5 years earlier, RVZ also induced potent humoral and cell-mediated immune responses [148]. Currently, the Advisory Committee on Immunization Practices (ACIP) recommends the administration of RZV to adults aged >50 years regardless of prior receipt of ZVL [149].

Although there are no current recommendations for booster vaccination, a cohort study of additional RZV vaccinations 10 years after the initial two RZV doses revealed a significant increase in both gE-specific antibody levels and gE-specific CD4+ T-cell frequency following booster vaccination [150]. Notably, the first booster vaccination led to an increase in immunity to levels even higher than those induced by the initial RZV vaccination. However, a single booster dose of RZV was sufficient because the second RZV booster dose did not increase the level of immunity to VZV beyond that induced by the first booster dose. The booster vaccination was well tolerated in this study, and no safety concerns were noted [150].

While ZVL is recommended for the immunization of individuals aged ≥50 years in most regions and has demonstrated efficacy, it is a live vaccine and is contraindicated for individuals with immunosuppression or immunodeficiency [151]. The RZV was the first vaccine approved for use in immunocompromised patients. The safety of RVZ in patients with autoimmune diseases and immunodeficiency (including individuals with hematologic malignancies, autologous hematopoietic stem cell transplantation, renal transplantation, solid tumors, and HIV infection) was found to be comparable to that in healthy individuals [152,153,154]. In patients with autoimmune diseases, the efficacy of RZV was relatively preserved compared with that observed in the general population [155,156]; however, it may be slightly reduced, particularly in those taking JAKis. Similarly, the immune response to RZV in immunodeficient or immunocompromised individuals was reduced compared to that in healthy individuals, particularly in hematopoietic stem cell transplant recipients, and the vaccine efficacy was reduced to approximately 68% at 1 year [153,154]. Nonetheless, vaccination remains effective in populations at high risk of HZ and its complications (Figure 4). The immune response to RVZ was considerably higher after the second dose than after the first dose, and this difference was more pronounced in older immunocompromised patients [152,153,154]. In 2022, the ACIP recommended two doses of RVZ vaccination for all immunodeficient or immunosuppressed individuals aged ≥19 years [157].

Numerous vaccine adjuvants used in vaccination have been designed to mimic TLR ligands. Consequently, these adjuvants, along with their supportive functions, have the potential to induce autoimmune diseases known as autoimmune/inflammatory syndrome induced by adjuvants (ASIA) [158]. Due to the presence of adjuvants such as AS01B in RZV, there is a theoretical risk that these adjuvants may induce or exacerbate autoimmune diseases. In a pooled population analysis of the ZOE-50/70 study, exacerbations or new-onset immune-mediated diseases due to vaccination were comparable between the RZV and placebo groups across all analyzed periods [159]. Potential immune-mediated diseases occurred in 0.7% of the patients, with gout, polymyalgia rheumatica, and psoriasis emerging as the most commonly documented conditions [160]. An analysis of pooled data from six clinical trials also indicated that new cases of immune-mediated diseases were comparable between the RZV and placebo groups. Furthermore, the incidence of immune-mediated diseases was not concentrated in any particular system organ class [161]. Findings from two single-center retrospective cohorts indicated that relapse of autoimmune disease following RVZ treatment was observed in 6.7–16% of patients. Among those with RA, relapses were documented in 5.0–24% of cases, a higher incidence than in patients with systemic rheumatic diseases. Most relapses are managed with GCs, whereas some necessitate a modification in immunosuppressive medication [162,163].

## 5. Conclusions

The incidence of HZ is increasing because of the growing population of older adults, patients with autoimmune diseases, and individuals with chronic diseases [8]. Patients with autoimmune diseases face a higher risk of developing HZ owing to the advent of drugs with new mechanisms of action that contribute to the public health burden [11,21,117]. In this era of immunosuppression, vaccination plays a crucial role in managing infections and their associated complications, consequently reducing pressures on public health resources [121,146]. However, the ZVL used thus far, despite multiple studies demonstrating its safety in immunocompromised patients, is generally contraindicated for this population owing to its live composition. Additionally, a challenge has been the reduced efficacy of the vaccine in immunocompromised patients compared to healthy individuals [134]. Hence, a vaccine designed to meet the specific requirements of immunocompromised populations is urgently required. A promising avenue is the emergence of the RVZ in recent years. RVZ is an inactivated vaccine approved for use in immunocompromised individuals [157]. Moreover, the adjuvant included in the RVZ can enhance the protection against HZ by augmenting the immune response, thereby compensating for the diminished vaccine response in immunocompromised individuals [142,144]. While adjuvants carry the potential risk of inducing new autoimmune diseases or flares of pre-existing autoimmune diseases, reports on such occurrences are limited [162,163]. It is essential to emphasize that this vaccine is safe and crucial for the prevention of HZ [158]. Drug safety is a critical component of any medical intervention, necessitating a careful evaluation of the risk–benefit balance. Our review provides valuable insights for clinical decision-making and public health practice.

## Figures and Tables

**Figure 1 biomolecules-14-00739-f001:**
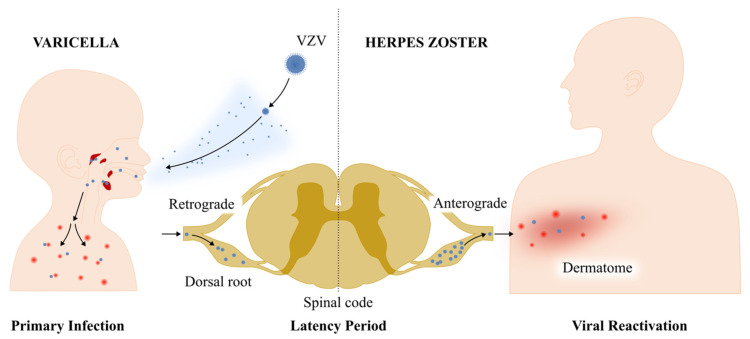
The life cycle of varicella-zoster virus. Varicella-zoster virus (VZV) infects humans when virions contact the respiratory mucosa of the host. The virus initially replicates locally and subsequently spreads to the tonsils and regional lymph nodes. It is then transported to the skin via T lymphocytes, causing a vesicular rash. During this period, the virion retrogradely travels to the nerve cell body of a sensory ganglion neuron from the site of skin replication, establishing a latent infection. Cellular immunity subsequently regulates VZV replication; however, with declining levels of cellular immunity due to aging or immunodeficiency, VZV reaches the skin through anterograde axonal transport, resulting in a vesicular rash on the skin segments innervated by the affected ganglia.

**Figure 2 biomolecules-14-00739-f002:**
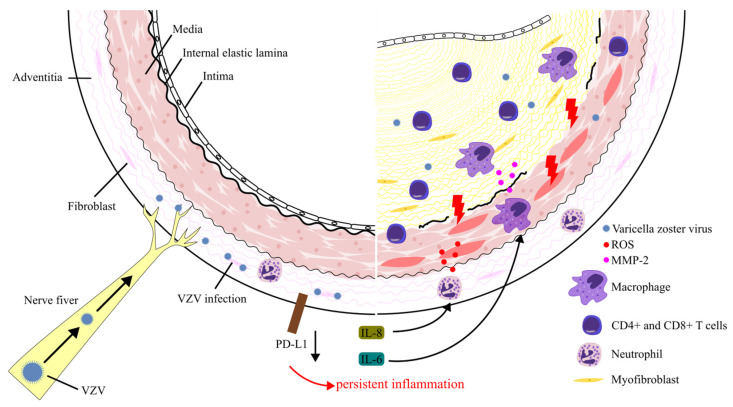
The potential mechanism of VZV vasculopathy. Varicella-zoster virus (VZV) reaches the vessel wall via anterograde axonal transport and infects arterial adventitial cells. VZV-infected adventitial cells show reduced expression of programmed death-ligand 1 (PD-L1) expression, which promotes persistent inflammation. Interleukin (IL)-6 and IL-8 levels are elevated in the cerebrospinal fluid of VZV-infected patients, facilitating neutrophil chemotaxis and macrophage differentiation in the vasculature. In addition, macrophages produce proteases such as matrix metalloproteinase (MMP)-2 and MMP-9, which disrupt the basement membrane and tight junctions between endothelial cells. These processes amplify vascular inflammation, leading to vascular damage and remodeling.

**Figure 3 biomolecules-14-00739-f003:**
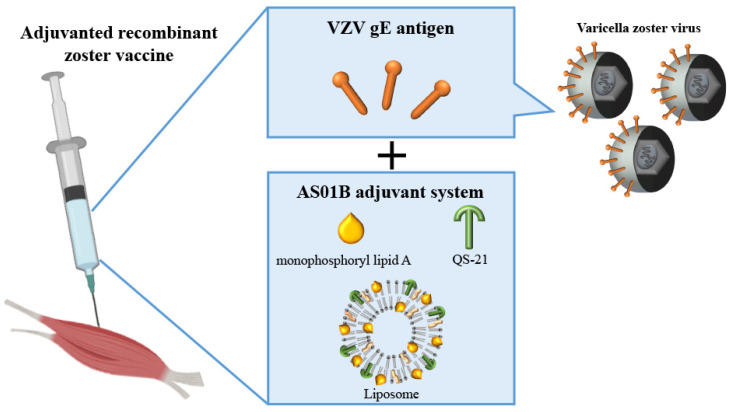
The effective compositions of the adjuvanted recombinant zoster vaccine. The adjuvanted recombinant zoster vaccine (RZV) is composed of the varicella-zoster virus (VZV) glycoprotein E (gE) antigen and liposome-based AS01B adjuvant. The VZV gE antigen is essential for virion assembly and viral replication, and it is recognized as a major target for VZV-specific CD4+ T-cell responses. The AS01B adjuvant contains two immune stimulants: monophosphoryl lipid A and QS-21. Monophosphoryl lipid A promotes nuclear factor-κB transcription and cytokine production. It works as a Toll-like receptor 4 agonist and stimulates antigen-presenting cells. QS-21 activates CD4+ T-cell-mediated immune responses and antibody production. The effective components of RZV can induce gE-specific CD4+ T-cell and humoral immune responses.

**Figure 4 biomolecules-14-00739-f004:**
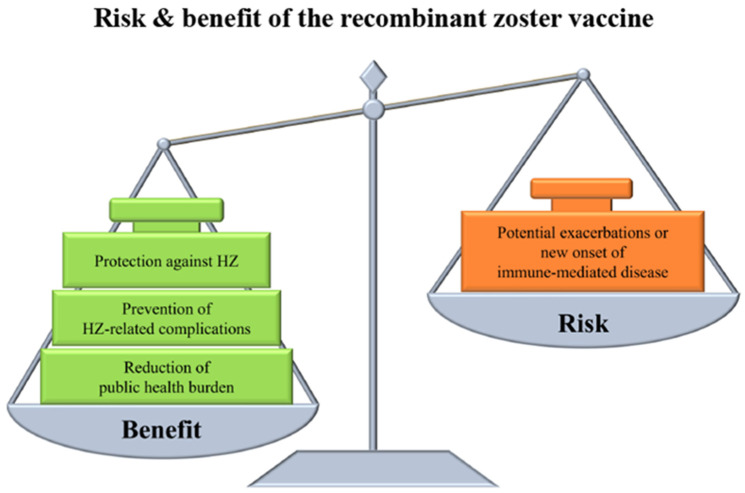
The risk–benefit balance of the adjuvanted recombinant zoster vaccine. The administration of the adjuvanted recombinant zoster vaccine (RZV) is associated with both benefits and risks. The sustained immune response induced by RZV provides long-term protection against herpes zoster (HZ). RZV can reduce the duration of HZ-associated pain. Furthermore, RZV has been demonstrated to be effective and safe for immunocompromised individuals and older adults. The benefits of RZV can mitigate individual complications and medical costs, potentially decreasing the public health burden. Conversely, there is a theoretical risk that AS01B adjuvants may induce autoimmune/inflammatory syndrome induced by adjuvants (ASIA). RZV may also exacerbate immune-mediated disease. Nevertheless, reports of these occurrences are limited. Given the risk–benefit balance, RZV can be considered an effective preventive measure, particularly for individuals at high risk for HZ and its associated complications.

## Data Availability

No new data were created or analyzed in this study. Data sharing is not applicable to this article.

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
