# Peer review of "Exploring the Link between Varicella-Zoster Virus, Autoimmune Diseases, and the Role of Recombinant Zoster Vaccine"

_biomolecules, 2024, doi:10.3390/biom14070739_

Round 1

Reviewer 1 Report

Comments and Suggestions for Authors

This article provides an overview of the association between varicella-zoster virus (VZV) and autoimmune diseases. It introduces the biological characteristics of VZV, the infection process, clinical manifestations, complications of varicella and herpes zoster, as well as their relationship with autoimmune diseases. The article offers detailed information on preventing herpes zoster in patients with autoimmune diseases, providing valuable insights for clinical decision-making. It analyzes the advantages and disadvantages of two main vaccines and delves into their safety and efficacy. The importance of preventive measures in reducing the public health burden, especially in immunocompromised patients, is emphasized.

However, there are areas for improvement:

1. The article may be somewhat lengthy, and excessive technical details could be simplified.

2. There is no discussion of new mechanisms of VZV replication and immune mechanisms, which are crucial for vaccine development.

3. The article does not address the epidemiological characteristics and disease burden of each autoimmune disease, which could help readers better understand the disease background and treatment needs.

4. There is a lack of description of post-vaccination clinical evaluations, which are crucial for understanding the long-term effects and safety of vaccines.

Author Response

Reviewer: 1

Comments to be transmitted to the Author
This article provides an overview of the association between varicella-zoster virus (VZV) and autoimmune diseases. It introduces the biological characteristics of VZV, the infection process, clinical manifestations, complications of varicella and herpes zoster, as well as their relationship with autoimmune diseases. The article offers detailed information on preventing herpes zoster in patients with autoimmune diseases, providing valuable insights for clinical decision-making. It analyzes the advantages and disadvantages of two main vaccines and delves into their safety and efficacy. The importance of preventive measures in reducing the public health burden, especially in immunocompromised patients, is emphasized.

Response: We thank the reviewer’s supportive and constructive comments. Based on your comments, we have revised our manuscript as below.

  1. The article may be somewhat lengthy, and excessive technical details could be simplified.

Response: We agree with the reviewer. We have shortened some technical details. However, we could not reduce the overall length of the article less than 4,000 words, which is required by the journal.

2. There is no discussion of new mechanisms of VZV replication and immune mechanisms, which are crucial for vaccine development.

Response: We agree with the reviewer. We emphasized that VZV viral gE is essential for the formation of infectious virions and viral replication, and further that gE is a major target for VZV-specific CD4+ T cell responses (Page 9, Line 390-392).

  1. The article does not address the epidemiological characteristics and disease burden of each autoimmune disease, which could help readers better understand the disease background and treatment needs.

Response: Thank you for your comment. We agree with the reviewer that epidemiological description is helpful for reads to understand. We have added epidemiological descriptions of each disease in the first paragraph of each (Page3, Line 110-114; Page 3, Line 135-139; Page 4, Line 161-165).

  1. There is a lack of description of post-vaccination clinical evaluations, which are crucial for understanding the long-term effects and safety of vaccines.

Response: We appreciate your suggestive comments. Based on the reviewer’s advice, we have described the vaccine efficacy at 10 years after RZV vaccination and the occurrence of adverse events based on the ZOE-50 and ZOE-70 trials (Page 10, Line 432-440).

Reviewer 2 Report

Comments and Suggestions for Authors

The authors in this review described the VZV infection in line with autoimmune diseases and discussed the role of recombinant Zoster vaccine. Case studies have been discussed to have more insights along with some pictorial representations. The review is comprehensive with minor revision required. 

1. Typing errors are there, for instance, line 169, please correct the word pathogenesis. Similar mistakes were found throughout the paper so please go through the text for grammar mistakes and spell checks.

2. Authors mentioned specific autoimmune diseases in connection to VZV, please clarify in the paper why the authors chose only those as described.

3. Line 251, authors discussed the role of type 1 interferons in Anifrolumab antibody. Can the authors discuss a little more into type 1 interferons specifically to get the detailed understanding of the link between type 1 interferons and the monoclonal antibody?

Comments on the Quality of English Language

minor revision required.

Author Response

Reviewer: 2

Comments to be transmitted to the Author
The authors in this review described the VZV infection in line with autoimmune diseases and discussed the role of recombinant Zoster vaccine. Case studies have been discussed to have more insights along with some pictorial representations. The review is comprehensive with minor revision required.

Response: We sincerely appreciate your positive comments.

1: Typing errors are there, for instance, line 169, please correct the word pathogenesis. Similar mistakes were found throughout the paper so please go through the text for grammar mistakes and spell checks.

Response: We appreciate your detailed check of our manuscript. We have checked the manuscript for grammatical and spelling errors.

2: Authors mentioned specific autoimmune diseases in connection to VZV, please clarify in the paper why the authors chose only those as described.

Response: Based on your comments, we have also described the association of VZV infection with other autoimmune diseases. However, we are sorry that it is not possible to describe every disease in detail (Page 2, Line 68-72).

3: Line 251, authors discussed the role of type 1 interferons in Anifrolumab antibody. Can the authors discuss a little more into type 1 interferons specifically to get the detailed understanding of the link between type 1 interferons and the monoclonal antibody?

Response: Based on your comments, we have added a description of type I IFNs and their receptors (Page 6, Line 274-280).

Reviewer 3 Report

Comments and Suggestions for Authors

Given an elevated risk of developing herpes zoster (HZ) in autoimmune diseases, the authors aimed to explore the link between varicella zoster virus (VZV) infection and autoimmune diseases, and discuss the benefits and risks of using recombinant zoster vaccine (RZV) in patients with autoimmune disease. Therefore, the present review provides clinical relevance for the associations of VZV reactivation with autoimmune diseases and the used immunosuppressive agents. This is an updated review with an inclusion of the newest findings. Some comments need to be addressed as follows:   

Major comments

1.      Numerous studies have reported the reactivation of herpes virus infections in COVID-19 pandemic, it would be better that the authors could present the link between VZV reactivation and COVID-19 infection or its vaccination.

2.      Patients with autoimmune diseases often present with comorbidities such as hypertension, obesity, diabetes, cardiovascular diseases, osteoporosis, and interstitial lung diseases. It would be helpful to report the associations of the comorbidities with HZ development in patients with autoimmune diseases.

3.      The risk of HZ varies in JAKi-treated patients of different ethnicities. In clinical trials, the incidence rate of HZ among tofacitinib-treated RA patients was higher in Japan and Korea (8.0 per 100 person-years and 8.4 per 100 person-years, respectively) than in Taiwan/China (3.0 per 100 person-years), or the global rate (4.0 per 100 person-years) [Arthritis Rheumatol 2017;69:1960-1968]. The authors would have the related descriptions regarding effects of ethnicity on the risk of HZ development in rheumatoid arthritis (RA) patients treated with JAKi.

4. The data regarding the impacts of prior HZ history on HZ risk after JAKi initiation in autoimmune diseases such as RA are limited. It would be interesting that the authors could present the related evidence and have some discussion of the underling mechanisms.

Comments on the Quality of English Language

No any comment

Author Response

Reviewer: 3

Comments to be transmitted to the Author
Given an elevated risk of developing herpes zoster (HZ) in autoimmune diseases, the authors aimed to explore the link between varicella zoster virus (VZV) infection and autoimmune diseases, and discuss the benefits and risks of using recombinant zoster vaccine (RZV) in patients with autoimmune disease. Therefore, the present review provides clinical relevance for the associations of VZV reactivation with autoimmune diseases and the used immunosuppressive agents. This is an updated review with an inclusion of the newest findings. Some comments need to be addressed as follows:

Response: We sincerely appreciate your positive comments.

1: Numerous studies have reported the reactivation of herpes virus infections in COVID-19 pandemic, it would be better that the authors could present the link between VZV reactivation and COVID-19 infection or its vaccination.

Response: Thank you for your comment. We have added a description of VZV reactivation after COVID-19 infection and vaccination (Page 2, Line 65-66).

2: Patients with autoimmune diseases often present with comorbidities such as hypertension, obesity, diabetes, cardiovascular diseases, osteoporosis, and interstitial lung diseases. It would be helpful to report the associations of the comorbidities with HZ development in patients with autoimmune diseases.

Response: Based on your comments, we have described the association between comorbidities and the development of HZ in patients with autoimmune diseases. However, we are sorry that the evidence is very limited (Page 3, Line 100-106).

3: The risk of HZ varies in JAKi-treated patients of different ethnicities. In clinical trials, the incidence rate of HZ among tofacitinib-treated RA patients was higher in Japan and Korea (8.0 per 100 person-years and 8.4 per 100 person-years, respectively) than in Taiwan/China (3.0 per 100 person-years), or the global rate (4.0 per 100 person-years) [Arthritis Rheumatol 2017;69:1960-1968]. The authors would have the related descriptions regarding effects of ethnicity on the risk of HZ development in rheumatoid arthritis (RA) patients treated with JAKi.

Response: We agree with the reviewer. We have described the effect of ethnicity on the risk of developing HZ (Page 7, Line 314-315).

4: The data regarding the impacts of prior HZ history on HZ risk after JAKi initiation in autoimmune diseases such as RA are limited. It would be interesting that the authors could present the related evidence and have some discussion of the underling mechanisms.

Response: Thank you for your comment. We have described the impacts of prior HZ history on HZ risk after JAKi initiation (Page 7, Line 313-315).